# Qualitative study exploring health care professionals' perceptions of providing rehabilitation for people with advanced dementia

Abigail J Hall [ID] ,[1] Fay Manning [ID] ,[2] Victoria Goodwin [ID] [1]

[1]Public Health and Sports Science, University of Exeter, Exeter, UK
[2]Department of Medical Imaging, University of Exeter Medical School, Exeter, UK

**Correspondence to**
Dr Abigail J Hall;
a.hall4@exeter.ac.uk

## ABSTRACT

**Objectives** The aim of this study was to explore healthcare professionals' principles for providing and delivering rehabilitation interventions for people with advanced dementia.

**Design** This was a qualitative study with three focus groups undertaken virtually. The data were analysed using a process of reflexive thematic analysis in order to gain an in-depth understanding of rehabilitation principles for this population.

**Setting and participants** 20 healthcare professionals who were specialists in treating and rehabilitating people with advanced dementia were recruited. These healthcare professionals had a wide range of experience in a variety of different settings including primary care, secondary care as well as specialist mental health teams. Purposive sampling focused on the requirement for participants to have significant experience of treating people with dementia. Participants were from the UK and Denmark. Data collection was undertaken during August and September 2022.

**Results** Three overarching themes were developed following analysis—organisational culture, knowledge and personal values of the healthcare professional. The first explored how the culture of an organisation affects a person with advanced dementia as well as the healthcare professional. The organisation needed to promote positive approaches to person-centred care and provide effective situational leadership to embed such approaches. Knowledge was a key consideration and was closely linked to the personal values of the healthcare professional, which formed the final theme. This study suggests that the interrelationship of these three factors influences the outcomes for the person with dementia and effective outcomes required consideration of all domains.

**Conclusions** Effective interventions for people with advanced dementia require the healthcare professional to have the knowledge about dementia and positive personal values, but the culture of the organisation is also key to ensure that the healthcare professional is able to deliver successful interventions.

## INTRODUCTION

Dementia is a neurological disorder resulting in physical and cognitive decline, found typically in older adults.[1 2] It is estimated that, by 2040, there will be 1.2 million people with dementia living in the UK—a 57% increase from 2016,[3] suggesting that dementia will become a national—and global—challenge to healthcare services. Advanced dementia is characterised by profound communication, memory and physical disturbances,[4] which often result in significant impairments in a person's ability to undertake activities of daily living and a perceived decline in quality of life.[5]

A variety of healthcare professionals (HCPs) can be involved in trying to reduce the physical declines associated with advanced dementia, including physiotherapists and occupational therapists. However, there remains little evidence to support physical interventions for people with dementia.[6–10] This lack of evidence is particularly evident for people with more severe dementia, where physical interventions to improve physical ability—or reduce the speed of decline—are largely not understood.[6]

### STRENGTHS AND LIMITATIONS OF THIS STUDY

⇒ This work was informed and supported by an active Patient Participation Involvement group—including collaborating with designing the study and discussing the findings.
⇒ The methods of data collection and analysis were robust and followed clear guidance for undertaking such qualitative work.
⇒ The study was reported using the Consolidated criteria for reporting qualitative research (COREQ) guidelines to ensure clear and transparent methods and analysis.
⇒ Although this was a small-scale qualitative study with a limited range of healthcare professionals taking part, which could impact on generalisability, sampling was deliberately specific and engaged healthcare professionals who were specialists in treating people with very advanced dementia.

There is much debate about the approach that is suitable for people with advanced dementia, with greater emphasis being placed on adopting a biopsychosocial model, where functioning and participation is key. Some 60 years ago, a key paper—'The need for a new model: A challenge for biomedicine'[11] argued that the dominant biomedical model of disease was insufficient for a complete understanding of health. It was stated that a biomedical approach "*assumes disease to be fully accounted for by deviations from the norm of measurable, biological variables. It leaves no room within its framework for the social, psychological and behavioural dimensions of illness*" (p130). Considering not only biological components but also the individual and societal contexts of the individual's experiences is vital in the rehabilitation of people living with dementia to ensure a person-centred approach to achieve what is meaningful to the individual. However, the challenge that this poses to HCPs is poorly understood—especially for people living with more advanced dementia.[6 12–14]

The experience of HCPs treating people with dementia has been explored in a variety of settings,[15 16] including residential aged care facilities[16 17] and acute settings.[18] A recent review exploring the experiences of caring for people with dementia in acute settings has highlighted the importance of developing knowledge of person-centred approaches to care; however, research has also highlighted a lack of understanding about the importance of the environment and how this may impact care.[19] However, evidence suggests that providing rehabilitation interventions for people with more advanced dementia is different to those with more mild forms.[6 14] Thus, the aims of the study were to explore the rehabilitation principles of HCPs providing and delivering rehabilitation interventions for people with advanced dementia in a variety of settings including acute, community and residential care.

## METHODS
### Design
During August and September 2022, three semistructured focus groups were undertaken with a range of HCPs. These health professionals were all specialists in the management or rehabilitation of people living with advanced dementia. The focus groups were undertaken virtually to reduce barriers due to geographical limitations and enabled international specialists to contribute. The study has been reported according to the Consolidated criteria for reporting qualitative research (COREQ) guidelines (online supplemental material 1).

### Setting and participants
A purposive sampling strategy was employed to seek HCPs who specialise in working with people with advanced dementia. A targeted social media advert sought professionals with advanced skills and pre-existing networks were drawn onto recruit participants. Some of these participants were known to the researchers in a professional capacity due to previous work undertaken in this area. This did not affect the data collection.

Inclusion criteria: specialist working with people with dementia as a physiotherapist, occupational therapist, psychologist, nurse or other related profession. ('Specialist' refers to the level of experience and training that an HCP has and an expectation that they are working in an advanced role relating to the care of people with dementia.)

Participants who responded to the social media advert or initial email were sent a follow-up email with the participant information sheet attached. At this point, they were offered an opportunity to discuss the study further if they had any particular questions. If they were happy to take part in the study, they were asked to provide informed consent prior to the commencement of the focus group. Participants were informed that they were under no obligation to participate in the focus group and they may withdraw from it at any time, without any negative consequence, up until the point where the data were fully anonymised.

Provided the participant met the inclusion criteria, all participants who consented to take part were invited to join a focus group. Recruitment continued until sufficient data were obtained to be able to answer the aim of the study and no new data were being generated.

### Data collection
The focus groups aimed to explore the challenges and techniques required to effectively manage people with more advanced dementia, while also considering the principles of rehabilitation for this population. The focus groups were facilitated by two experienced qualitative researchers (AJH and FM). FM acted as a 'second facilitator/observer', taking notes and making observations pertaining to interaction of the group. Participants were all encouraged to give their opinions and thoughts during the meeting. To ensure that everybody was given the opportunity to contribute, each participant was asked whether they had further detail to add to any of the topics before moving onto the next. Both researchers have undertaken training courses pertaining to qualitative analysis and have published multiple qualitative papers.

Microsoft Teams was used to undertake the focus groups and they were recorded and automatically transcribed. The transcription function has been used by the researcher and found to be accurate, although the transcripts were all checked for accuracy. The focus groups lasted for a maximum of 90 min and included time for participants to introduce themselves and their roles to other members of the group. The participants were informed that this study formed part of a larger piece of work aiming to develop interventions for people with advanced dementia. A semistructured guide (online supplemental material 2) was used to guide the conversation of the focus groups. The topic guide was not piloted prior to the initial focus group as it was anticipated that it could be adapted for subsequent focus groups if needed.

This was not required and the same topic guide was used for all interviews. The data were anonymised, so that individual participants could not be identified, and all participants consented to their data being transcribed and analysed. Transcripts were not reviewed by participants. The data collection from the focus groups provided rich and detailed data and no further follow-up interviews were deemed necessary although people were offered the opportunity to discuss any further points following completion of the focus group.

### Patient and public involvement

A patient and public involvement (PPI) group was developed to help inform the research undertaken during this study and as part of a wider programme of work. The group has been evolving and developing and now consists of people with a variety of experiences of dementia. The group includes people living with dementia, carers of people with dementia and two HCP who have personal experience of dementia.

Initial discussions focused on the challenges they had experienced with getting rehabilitation for people who have dementia—especially when the person had advanced dementia or was living in a care home. Several themes generated from the PPI members formed part of the focus group discussion. Due to practical challenges, none of the PPI group was able to participate in the focus groups, but discussion was held with representatives following the groups and helped guide the analysis and results.

### Analysis

Analysis commenced after the first focus group was completed and continued throughout the following focus groups. As the data collection and initial analysis ran simultaneously, arising themes or gaps could be probed accordingly in the remaining focus groups.

NVivo V.11 (QSR International) was used to organise and code the data. This facilitated a process of reflexive thematic analysis to be undertaken—as suggested by Braun and Clarke.[20] Familiarisation of the data involved reading and rereading the transcripts. A process of systematic open coding was then commenced and initial ideas for themes were noted. Following this open coding, initial themes were generated, discussed and explored with the wider research team. These discussions led to further refinement of the themes, and definitions were written to ensure clarity between the research team. For example, there were multiple codes relating to participants using positive success stories to educate others, this was collated into a minor theme of 'optimism/wisdom,' which was then classified within the 'personal values of the HCP'. This process was repeated for all the data until themes were developed. This process was an iterative process and resulted in several refinements of named themes. These themes were then discussed with the PPI group who were supporting and collaborating on the overarching project.

## FINDINGS

Twenty HCPs took part in three focus groups (table 1). Only one person who had initially expressed an interest in taking part did not reply to the meeting invite and, therefore, did not take part. Following the third focus group, it was felt that no new themes were occurring and, therefore, it was believed that theoretical data saturation had been reached and no further focus groups were planned. There were no withdrawals from the study at any stages. The HCPs reported an average of 18 years of experience and an average of 13 years specialising specifically with people with dementia. The settings of their experience were broad, including in-patient settings, out-patient clinics, patients' own homes and residential/nursing settings.

The principles of rehabilitation were explored for people with advanced dementia and three overarching themes were developed following analysis. These were 'organisational culture,' 'knowledge about dementia' and 'personal values of the HCPs'. The first theme explores how organisational culture may affect the person with advanced dementia as well as the HCP working within a system or organisation. Second, knowledge of dementia was considered to be a key consideration when exploring the rehabilitation of people with advanced dementia, but this was closely linked to the personal values of the HCP, which formed the final theme. Each theme will be explored in further detail and then the interaction between them considered.

### Organisational culture

Our participants discussed how there was a varied understanding of what 'rehabilitation' means to people and how this differing understanding led to people with advanced dementia receiving a wide variety of interventions. This often related to understanding what the purpose of rehabilitation was, from seeking to gain functional improvements to preventing decline. There was also discussion about the consideration of how services and organisations' cultures affected the interventions that were delivered.

Determining what the term 'rehabilitation' means was reported to be challenging for HCPs generally. What an organisation considered to be rehabilitation would significantly affect what was delivered. Our participants were very clear that for people with advanced dementia, tasks that were perceived to be 'simple' were actually very important and, although not deemed as 'traditional rehabilitation' were actually important aspects of rehabilitation.

> we talk a lot in our training about the concept of rehab as what we find is if…. they're not looking to get them back up on their feet, then that's not rehab. Whereas my argument that is actually for somebody with quite an advanced dementia, we have help them into a sitting position, can have massive functional benefits for them. So that in itself is rehab (PT9)

| Table 1 | Participant characteristics (n=20) |
|---|---|
| **Sex** | |
| Female | 19 |
| Male | 1 |
| **Profession** | |
| Physiotherapist | 14 |
| Occupational therapist | 3 |
| Nurse | 2 |
| Psychologist | 1 |
| **NHS grade/banding** | |
| 6 | 6 |
| 7 | 8 |
| 8+ | 2 |
| N/A | 4 |
| **Years of clinical experience** | |
| 0–5 | 1 |
| 6–10 | 2 |
| 11–15 | 2 |
| 16–20 | 3 |
| 21–25 | 6 |
| 26–30 | 2 |
| 30+ | 3 |
| Not reported | 1 |
| **Years of experience working with people with advanced dementia** | |
| 0–5 | 4 |
| 6–10 | 2 |
| 11–15 | 6 |
| 16–20 | 5 |
| 21–25 | 2 |
| 26–30 | 1 |
| 30+ | 0 |
| **Location** | |
| England | 13 |
| Wales | 2 |
| Scotland | 4 |
| Denmark | 1 |
| **% of caseload of people with dementia** | |
| 0–25 | 1 |
| 26–50 | 1 |
| 51–75 | 6 |
| 76–100 | 8 |
| Not currently working clinically | 4 |
| **Locations of work** | |
| In-patient | 12 |
| Out-patient | 4 |
| Care home/residential homes | 11 |
| | Continued |

| Table 1 | Continued |
|---|---|
| Own homes | 11 |
| Other | 2 |
| NHS = National Health Service | |

The term 'traditional rehabilitation' was discussed in all focus groups and led back to the underlying biomedical principles that historically have been adopted by many HCPs. This approach was focused on the injury or illness.

> ……[we] know all about anatomy and physiology and this is how to treat a broken leg. This is how to treat a back pain. But they don't actually get taught how to embrace that full person centered approach to things (PT7).

There was also a noted discomfort that HCPs felt that they had to justify doing things differently—and, thus, outcome measures were described as being vital to 'prove effectiveness' of their interventions. Often organisations demanded these outcome measures to prove that their services were effective. It was also discussed that often it was only the physical interventions that were perceived to be treatment—to such an extent that only the physical interventions were often documented. Our participants were very clear that these 'non-traditional' approaches needed to be clearly documented to justify their interventions.

> I think is well, it's about how we document that because we don't always document those times when we've gone in and not been able to engage with somebody. We maybe say they weren't able to engage in our documentation, but we don't say we spent time engaging with that person and it resulted in an improved rapport, improved connection (PT7).

Complexity of patients was noted as requiring a different approach and people with advanced dementia were recognised as being highly complex and, therefore, needing care to be adapted accordingly.

> We're trying to pretend that we haven't got a complex system….which a complex patient is…. we're still trying to imagine that we've got a simple machine where if you put one thing in change, one of the dials, you're gonna get a completely different product out. And so it goes right back to undergraduate education and absolutely rethinking our model of physiotherapy (PT1)

It was discussed that the focus of rehabilitation has often been to make a tangible improvement after an illness or injury, which initially resulted in a decline in physical function. However, our participants questioned this and reported that often for people with advanced dementia, rehabilitation could be considered a slowing

of the inevitable physical decline that is associated with the condition.

> ….because it's a degenerative disorder and …. the processes that these patients will deteriorate……it's not just about improving things for them, but maintaining what they're able to do for as long as physically possible as well (PT6)

While there was a focus on preventing decline, there was also belief that people with advanced dementia can improve physically—therefore, this should be a focus of the intervention. However, it was clear that this potential to improve physically required timely—and appropriately targeted—interventions, often before there was significant physical decline.

The unclear nature of interventions for people with advanced dementia makes it difficult to determine their trajectory of physical recovery; however, our HCPs felt very strongly that people needed to be given an opportunity to receive rehabilitation. In the acute setting, giving these opportunities was challenging due to the focus on discharging patients to ensure good patient 'flow'. Despite this challenge, our participants were determined to offer as much as they possibly could, but they recognised this was often insufficient.

> Yeah, time, time for us… how long you can spend with the patient. Obviously you know you have so many patients to split your time between. But our biggest time pressure is discharge pressures (PT13).

The influence of the service or organisation that a person was working within had a significant impact on the interventions that could be delivered. The way that an organisation did, or in some cases did not, adapt their environments and policies to suit a person with advanced dementia would influence their journeys.

> The acute sector is massively, massively limited in what it can provide people with dementia…. the environment is completely wrong. It's wrong for people without dementia. It's completely wrong for people with dementia, so of course they're not gonna give their best selves (PT4)

While the acute setting had a significant focus on patient flow, the actual environment was deemed very unsuitable for people with advanced dementia. Noise, inappropriate lighting and distractions would often add to challenging behaviours that the person might display. Organisations needed to ensure their environments were suitable.

> we do seem to have a bit of an issue, I call it beige bays, we create these spaces and we set people in them and we say 'now you stay there for six hours, 10 hours, 12 hours and only get up if you need to go to the toilet'. We….actively deprive people of activity during their stay and then get really shocked why they start getting really curious/bored out of their brains and wanting to get up and moving (OT2).

Organisations often were risk averse and fear of people falling and injuring themselves led to risk averse behaviours such as discouraging mobility or keeping people in bed, but HCPs feared this and actively tried to overcome these challenges. Discouraging activity led to deconditioning and deterioration of function which was considered a significant risk of harm in itself.

> I don't like the term deconditioning, because I think it's completely poor care and lack of facilitating to function… We've allowed this this appalling situation to happen where when our older people go into hospital, they deteriorate almost in front of in fact definitely in front of our eyes (PT1)

There was a frustration that often people with advanced dementia were restricted from undertaking 'risky' activities, but there was little emphasis placed on how this would affect the person from an emotional perspective.

> I hear people talking about the talking about the risk of falling over the risk of pressure ulcers, that risk of infection. We have huge risks to personhood, to selfhood, to self esteem, to independence, to the ability to live independently (PT1)

Our HCPs also reported frustration with arbitrary timescales that patients were given to receive an intervention. Whether this was due to waiting lists for community services, which were reported to be up to 40 weeks in some services, to the time limited services such as in-patient intermediate care or rehabilitation.

> ….if you're gonna give people time, you need to look at the whole system, that of rehab….who came up with six weeks? I've got no idea. But that seems to be "if you can't get it and six weeks, then there's no point really" attitude (PT9).

### Knowledge relating to dementia

Knowledge relating to dementia and a good understanding of person-centred care were deemed vital to be able to effectively treat a person with advanced dementia. Where there was a lack of knowledge, there were often poor outcomes for the person.

Knowledge relating to dementia was seen as being key to providing effective interventions. Our HCPs described challenges in gaining this knowledge, with many relying on experience and learning 'on the job'. Undergraduate education for HCPs was universally felt to be inadequate and therefore, postgraduate education was vital, although difficult to access.

> The standard of university education on dementia for physios in particular and frailty is really, really poor and there's a lot and a lot of education that us more experienced physios need to be doing…. and that's why the words "no rehab potential" ….[are] still out there. Because there's not a lot of education around frailty and older people (PT4)

HCPs working in mental health settings found access to specific mental health training more accessible generally, although this was often generic rather than profession specific—this was particularly reported by physiotherapists who reported a lack of physiotherapy_specific dementia training. There was a universal feeling among our participants that a large element of their role was about teaching and educating others—about dementia itself, and also the importance of delivering person-centred care approaches to interventions. This is not only related to other HCPs but also to patients themselves and their carers.

A lack of knowledge and understanding was deemed to precipitate the belief that people with dementia had 'no rehabilitation potential'. Indeed, this term was reported to be used very commonly.

> the staff who are labelling individuals of "no rehab potential" are often the less experienced staff …… making a snapshot sort of decision within 10 minutes or five minutes. (PT2)

Person-centred approaches to care were discussed in relation to a person with advanced dementia and also in terms of how treatments should be adapted and personalised for everybody.

> It doesn't really matter if a patient has advanced dementia or if they're me, a 28 year old that's had knee surgery. If an exercise makes no sense to me, or if it's boring, or if it's not gonna help me get back to what I want to do, I'm probably not gonna do it. And really, if we make healthcare better for those with advanced dementia, we're gonna make it better for everyone, because what we're talking about there is personalized care, isn't it? (PT4).

This emphasis on ensuring that interventions were tailored to individuals was reported to be even more important to a person with advanced dementia, but understanding how to tailor interventions was variable and influenced by external factors such as time and availability of resources.

> …that time pressure because I think patients feel it and I think I might, you know, [allied health profession] colleagues feel it as well because literally as soon as people are either hitting a rehab ward or getting into an [intermediate care] bed, you're looking at discharge planning before you've even really got to the crux of that rehab (PT9).

Our HCPs reported that people were very aware of the concepts of person-centred approaches to care, but these were often seen as gold standard and their services were not always deemed able to achieve this. Nor did they always have a good understanding about how to actually deliver such approaches. There was also a suggestion that person-centred care was not taught at undergraduate level in some professional groups, and this was something that had to be learnt as a post-graduate.

## Personal values of the HCP

It was evident that the personal values of HCPs providing care for people with advanced dementia could significantly impact on the delivery of that care.

An interest in, and taking time, to get to know a person was deemed vital to be able to provide interventions to people with advanced dementia. However, a natural curiosity was evident among our participants. Where little formal education had been received, their curiosity to discover 'what works' led them to a particular interest and passion for treating people with dementia. However, this was deemed to be the exception rather than the rule, and many of our participants reported feeling like a '*lone voice*' where few other people had an interest in this population.

> And the other point I just want to make is sometimes you feel like a lone voice out here kind of shouting about how important these things are. And I feel like I'm not on my own for a while (PT7)

However, not all HCPs had this level of curiosity or interest about working with people with advanced dementia. Many of our participants provided teaching and education to others relating to dementia, but often with little positive engagement.

> I actually got onto the physio course for them for me to teach them and I did two consecutive years …… People weren't interested…… a room of 50 people just shut down, as if cognition and old and people with cognitive frailty was something they'd never come across (PT1).

This was particularly reported among physiotherapists and there was a belief that occupational therapists were more involved in treating people with dementia or cognitive difficulties historically, although there was a feeling that this attitude was beginning to change. There was further evidence to support this with our participants discussing the role of physiotherapists in mental health teams being much less advanced than the role of occupational therapists.

> I was one of the first physiotherapist who was hired to work in a nursing home and in [country]……I've worked a lot with nurses and caregivers and they've been really interested in what I could offer. I have [tried with] the physiotherapist. They weren't interested (PT5)

There was considerable optimism among our participants that people with advanced dementia could effectively receive rehabilitation and other interventions, but this came from positive previous experience—their wisdom. It was notable that sharing this experience and positive outcomes was used as a way to engage other HCPs in treating people with advanced dementia.

Knowledge of negative outcomes for people with dementia also led dementia specialist HCPs to be more proactive in their approaches, often actively seeking out patients who would benefit from specialist interventions.

….we've also changed our practice. Whenever anyone go to the to that trust, we've phoned and we make sure and we may actually go out (PT10).

However, this optimism was often diminished by challenges relating to service pressures and limitation that the HCPs had to work within. Unsuitable ward environments were a particular challenge and the HCPs accepted the fact that the person with advanced dementia was unlikely to be able to improve in that setting and, therefore, focused their attention on ensuring they could be discharged to the most appropriate location to facilitate rehabilitation. Where this optimism was reduced, HCPs noted feeling very alone and isolated when trying to 'fight' for people with dementia.

It was evident that our participants felt a significant amount of compassion towards their patients. There was discussion about the mutual upset that they felt when people with dementia received suboptimal care, and this was clearly upsetting to participants—some giving examples of poor care that they had experienced. There was a belief that people with advanced dementia were often stigmatised and labelled, which prevented them from receiving the care that they needed and deserved.

…there's this been this historic thing about 'no rehab potential'. And the consistent negative labelling of our people that we work with…….'No capacity, no rehab potential behaviours that challenge aggressive.' That that's always been the negative, negative and pejorative labelling (PT1).

There was also a feeling of guilt and frustration where an HCP was not able to provide the intervention they wanted to—usually due to resource pressures or the culture of the organisation. However, it was evident that the individuals' attitude was key and their beliefs and engagement with people with advanced dementia had a pivotal role in determining what intervention was delivered.

….attitude is huge. I think if you if you don't have the right attitude, nothing really follows…… three words that stand out for me are time, connection and value. So value in the person value, in the carer value, in their journey and where they've got to until now (PT7).

The inter-relationship between the values of the HCP, the organisational culture and the knowledge of the HCP is detailed in figure 1. Our data suggest that in order for the person with dementia to have positive outcomes, all three variables need to be optimised.

## DISCUSSION

The aim of the study was to explore the rehabilitation principles when providing and delivering rehabilitation for people with advanced dementia. Our participants described the key aspects to consider the culture of rehabilitation and raised questions about not only what this means but also what other people and HCPs perceive it to be.

Our data suggest an important relationship between these factors and for a person with advanced dementia to receive optimal interventions, there needs to be a symbiotic relationship between all three factors (figure 1). Where any one of the elements is missing, the person with dementia will not receive the care that they need. However, there was a noted importance on the values of the HCP themselves and how these values could be directly influenced by the culture of the organisation and the knowledge about how best to manage people with advanced dementia.

There was a common belief that knowledge about treating people with advanced dementia was key and where there was a lack of knowledge outcomes were poor—or people simply were not offered interventions. This knowledge related both not only to the pathogenesis of dementia and how to provide care for the person but also to how interventions could be suitably tailored to accommodate the persons' cognitive abilities.

Knowledge about dementia among HCPs is inconsistent, with one large study undertaken by Alzheimer's Disease International suggesting that 62% of HCPs believed that dementia is a natural part of ageing and supported other studies' findings that there was nothing that could be done to manage it.[21 22] However, it is unclear whether, and how, this relates to specific professions. Our data suggest a lack of undergraduate training for physiotherapists relating to dementia. A Canadian study surveyed physiotherapists and found that levels of knowledge relating to dementia were high but cited lack of confidence to manage behavioural and cognitive difficulties.[23] Interestingly, similar research exploring the attitudes of physiotherapy students found that 53% felt that their academic training was sufficient to equip them with the skills to work with people with dementia,[24] suggesting a reduction in confidence from undergraduate level to postgraduate level.

Our data suggest that the personal values, and, thus, the attitude, towards a person with advanced dementia was pivotal in determining the care that was delivered and received. Our participants were all specialists in treating people with advanced dementia and, therefore, had compassion and interest in treating this population; however, they reported numerous examples where poor personal values of the HCP negatively affected the person with dementia—and, thus, their outcomes. Research has explored the importance of providing education and training for HCPs relating to dementia in the acute setting,[25] with a highlighted requisite that this education supports the development of empathy and understanding relating to the person with dementia.[25 26] However, much of this research focuses on attitudes in relation to knowledge, rather than the personal values of the HCP and how these influence the care provided.

| Values of the HCP | Organisational Culture | Knowledge of the HCP | Outcomes | |
|---|---|---|---|---|
| | | | For the person living with dementia | For the HCP |
| Good | Good | Good | Positive outcomes | Satisfaction and ability to make a positive improvement in care |
| Good | Good | Poor | Moderate outcomes | Willingness and desire to improve outcomes, but insufficient knowledge to achieve good outcomes |
| Good | Poor | Poor | Moderate outcomes | Frustration that the HCP values the person with dementia but has neither the positive culture within the organisation or the knowledge to make a positive change. |
| Good | Poor | Good | Moderate outcomes | Frustration of HCP being unable to achieve good outcomes for the patient as the organisation does not support their interventions |
| Poor | Poor | Good | Moderate outcomes | Unable to use the knowledge they have in an effective manner |
| Poor | Poor | Poor | Poor outcomes | Lack of interest or concern from HCP. Skills and knowledge deteriorate further.[i] |

**Figure 1** Theoretical model detailing the interaction of factors and the outcomes for the person with dementia and the HCP. Green denotes positive outcomes, yellow denotes moderate outcomes and red denotes poor outcomes. HCP, healthcare professional.

The culture of the organisation was suggested to influence the HCPs values and conversely, HCPs with poor values relating to dementia could negatively affect the culture of the organisation. The culture of care has been linked to the value placed on the person with dementia,[27] thus placing greater value on the person facilitates a culture which supports them. However, it has been suggested that education alone is not sufficient to facilitate this positive culture.[25 26] Organisations need a strong, positive leadership culture, which promotes person-centred care[28] and such leaders need to act as role models to demonstrate positive care. For people with complex needs, such as those with advanced dementia, situational leadership is recommended in the nursing literature[29] to support those with less experience. Lack of time and suitable environments were cited as being major barriers to facilitating effective interventions

for people with advanced dementia and must be considered key to developing this positive culture.

Our data suggest that the inter-relationship between the knowledge and values of the HCP and the culture of the organisation are all key to successful outcomes. Our data (figure 1) suggest that only positive attributes of all three domains will result in successful outcomes for people with advanced dementia.

This study used robust methods of data collection and analysis and followed clear guidance for undertaking such qualitative work. This included the use of COREQ (online supplemental material 1) to ensure clear and transparent methods and analysis. Although this was a small-scale qualitative study with a limited range of HCPs taking part—which could impact on generalisability—sampling was deliberately specific and engaged HCPs who were specialists in treating people with very advanced dementia. This reduced

the potential sample size, as there is a limited sample of people due to the specialist nature of interventions.

## CONCLUSION

This qualitative study contributes to the knowledge about key domains to consider when providing rehabilitation interventions for people with advanced dementia. We propose that there are three key principles which must be optimised to ensure the best outcomes for people. The organisational culture should support person-centred care approaches, while providing sufficient resources to support interventions. The HCP must have appropriate dementia-specific knowledge and their personal values must align with the requirements for people with dementia. Should all key domains be optimised, the person with advanced dementia has potential to engage successfully with rehabilitation interventions. Further research is needed to determine the content of the actual interventions that are delivered; however, we recommend that considerations are made to ensure that the person with dementia has the best chance of receiving meaningful rehabilitation.

**Acknowledgements** The authors thank all the participants who devoted their time to participate in this study.

**Contributors** Conceptualisation: AJH; methodology and responsibility for overall content : AJH, FM, VG; formal analysis: AJH, FM, VG; data curation; AJH, FM, VG; writing—original draft: AJH; writing—review and editing: AJH, FM, VG; supervision: VG; all authors have read and agreed to the published version of the manuscript.

**Funding** This study is independent research funded by the National Institute for Health and Care Research (NIHR) School for Primary Care Research (119247) and supported by the NIHR Applied Research Collaboration South West Peninsula. The views expressed in this publication are those of the author(s) and not necessarily those of the National Institute for Health Research or the Department of Health and Social Care.

**Competing interests** None declared.

**Patient and public involvement** Patients and/or the public were involved in the design, or conduct, or reporting, or dissemination plans of this research. Refer to the Methods section for further details.

**Patient consent for publication** Not applicable.

**Ethics approval** This study involves human participants and was approved by University of Exeter's College of Medicine and Health Research Ethics Committee - reference 515745 Participants gave informed consent to participate in the study before taking part.

**Provenance and peer review** Not commissioned; externally peer reviewed.

**Data availability statement** Data are available upon reasonable request. The datasets generated during and/or analysed during the current study are available from the corresponding author upon reasonable request.

**ORCID iDs**
Abigail J Hall http://orcid.org/0000-0002-3453-5631
Fay Manning http://orcid.org/0000-0002-9768-1695
Victoria Goodwin http://orcid.org/0000-0003-3860-9607

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
