## [Reviewer comments · BMJ Open]

ARTICLE DETAILS

TITLE (PROVISIONAL)	A qualitative study exploring health care professionals' perceptions of providing rehabilitation for people with advanced dementia
AUTHORS	Hall, Abigail J.; Manning, Fay; Goodwin, Victoria

VERSION 1 – REVIEW

REVIEWER	Anthony Scerri Department of Nursing, Faculty of Health Sciences , University of Malta
REVIEW RETURNED	24-Apr-2023

GENERAL COMMENTS	This is a very interesting article on an important topic. Overall the manuscript is well written. These are some suggestions to improve it further. My main suggestion is to describe more clearly what you mean by 'specialists' when you refer to the participants who were included in the study. This should be further explained. Other suggestions are the following: Abstract: Objectives: Can you clarify what you mean by theoretical basis? Background: In the background you can include similar studies that have looked at the opinion/perceptions of healthcare professionals towards dementia rehabilitation (for example: https://onlinelibrary.wiley.com/doi/abs/10.1111/jocn.16635) and explain more clearly why this study is different from the rest. Page 4 – Line 37 – Same as above? What do you mean by theoretical basis? Can you clarify and elaborate more on this? Page 5: Data collection: Considering that the focus groups were done online, how did the authors ensure that the voices of all participants was heard? Page 6: Patient/Public involvement: Can you clarify exactly who were the persons consulted? Were they informal caregivers, associations/societies or persons living with dementia? Page 6 Line 24: Typo: Held not had Findings Can the quotes be slightly tweaked to be more clear? For example by removing repeated words? Page 8: Theme: Culture: Would the term 'organisational culture' be more fitting here? Page 10 Line 45-48: "The influence of the service or organisation
---

	that a person was working within, or the person with advanced dementia was being treated by, had a significant impact on the interventions that could be delivered” – This sentence is not clear. Consider rephrasing. Page 11 : Sub-theme ‘Dementia’ could be rephrased ‘dementia knowledge’ Discussion: Page 15: Line 24: Table 2 not Table 1 Page 16: Line 6: You rightly argued that training/education alone is not enough to change the organisational culture. Can you elaborate more on factors that can support this change in culture (e.g., effective leadership, more resources, person-centred vision etc.) Consider including the limitations of the study. Also consider discussing some recommendations emanating from the study.
--	---

REVIEWER	Miia Rahja Flinders University
REVIEW RETURNED	02-May-2023

GENERAL COMMENTS	This qualitative study explores health care professionals’ thoughts about rehabilitation to people living with advance level dementia. The topic is interesting and the paper has potential to capture many readers, but there are some major edits that need to be done before I would recommend this for publication. The introduction does not give a clear justification for this study. The mention of this study providing a theoretical basis is ambitious and somewhat misleading and I would suggest removing any reference to this. Several of the quotations provided are oddly worded and I wonder if this is because the participants were from non-English speaking background or the transcription has not been properly checked. Abstract  - Missing detail about what methods exactly were used. It is also odd that strengths and limitations are listed in here. - This paper has added to the literature about HCP thoughts/knowledge/beliefs/attitudes (whatever you want to call it) about providing care for people living with advanced dementia but with a sample size of 20, it does not provide a theoretical basis. - The results in this sections should be reworked as well. Background. This section is very short and does not give clear justification to why this study is needed. My initial thought was that perhaps some of the theories that have been used to understand rehabilitation should be discussed, and why they may not suit in the context of advanced dementia. However, after reading your paper, I would suggest you remove any references to theory, or theoretical basis. This paper provides results from a qualitative study that add to other literature on the same topic and has the potential to be an interesting read. There have been other studies that discuss interviews with healthcare professionals about rehabilitation for people living with dementia, perhaps that should be the focus and how this paper is different to those studies and why this study was needed. Methods  - Remove section about PPIE on page 5 as you also have this as sub-heading and section on page 6 - Data analysis section could give example of the narrowing of the codes to themes that are presented in this paper.
---

	Results  - Table 1 – given “other’ is just 1, could you name this profession as well. - Table 1 – how can years of clinical experience be n/a if everyone was a HCP? - Same for % of caseload of people with dementia – maybe these should be explained in a footnote. - Page 9, line 23 – when you state “several” can you add in brackets (n=?) so this is clear to readers – I would do this to all such statements. - Page 9 line 30 – check capitalisation - Page 10, line 31, - consider revising sentence beginning with: Where the acute setting was often... - Also not sure the end of this section is about determining purpose. - Page 11, line 5 – poor lighting or inappropriate lighting ? - Page 12 line 34 – is there a typo in this quote - Page 13, lines 55-59 – again this quote seems a bit odd as it’s meaning is lost with poor grammar, for example, what does ‘I have [with] the physiotherapist’ mean or refer to? - The use of subheadings under each theme makes reading of the results “clunky”. Are they necessary? Rather, focus on presenting the story that makes up each theme. - Page 14, line 35 - How was it evident that your participant felt significant amount of compassion towards their patients? Discussion  - Page 15, line 2 should this refer to Table 2? - Page 15, line 27, how exactly does the data suggest that “knowledge is a key domain to ensure people with advanced dementia receive optimal care”? - Page 15 line 45 I suggest you remove terms such as “it was very evident” – it might be better to err on the side of caution especially as the paper has now clearly shown how many participants said such thing. - It is not quite clear how you came in to the conclusion about some of the outcomes presented in Table 2. Also, how do you know that if all three “themes” from your study are poor the outcome for person with advanced dementia is “Vicious spiral of worsening outcomes”. I also wonder if (a revised version of this table – if you choose to keep it) would be better at the end of results with a summary paragraph that clearly explains it. It would be better to focus the discussion on positioning you study among other similar studies. - Study strengths and limitations are missing, should these be moved from the abstract to here? Conclusion Consider revising, would be it safer to state this study may help guide future practice in care of people with advanced dementia?
--	---

VERSION 1 – AUTHOR RESPONSE

Reviewer: 1 Dr. Anthony Scerri, Department of Nursing, Faculty of Health Sciences , University of Malta Comments to the Author: This is a very interesting article on an important topic. Overall the manuscript is well written. These are some suggestions to improve it further. My main suggestion is to describe more clearly what you mean by 'specialists' when you refer to the participants who were included in the study. This should be further explained. Other suggestions are the following:	Thank you for taking the time to review this paper. We are grateful for your comments and have reworked the manuscript according to your comments. Clarification has been added under the “setting and participant” section to explain more the
---	---

	classification of “specialist”
Abstract: Objectives: Can you clarify what you mean by theoretical basis?	This is a good point. We agree that we didn’t explore the theoretical basis – more so the principles of rehabilitation. We have reworded this throughout the paper to clarify this point.
Background: In the background you can include similar studies that have looked at the opinion/perceptions of healthcare professionals towards dementia rehabilitation (for example: https://eur03.safelinks.protection.outlook.com/?url=https%3A%2F%2Fonlinelibrary.wiley.com%2Fdoi%2Fabs%2F10.1111%2Fjocn.16635&data=05%7C01%7Ca.hall4%40exeter.ac.uk%7C7aa2131c1c534633da1008db4bda19d6%7C912a5d77fb984eeef321334d8f04a53%7C0%7C0%7C638187172329048346%7CUnknown%7CTWFpbGZsb3d8eyJWljoicjoiMC4wLjAwMDAiLCJQIjoiV2luMzliLCJBTiil6lk1haWwiLCJXVCi6Mn0%3D%7C3000%7C%7C%7C&sdata=6%2Fh3kgJ%2BSEND9OUxMRtclDeyDIgyAln6SzHFEMlrvSg%3D&reserved=0) and explain more clearly why this study is different from the rest.	We have added further detail and rationale for this study in relation to existing evidence
Page 4 – Line 37 – Same as above? What do you mean by theoretical basis? Can you clarify and elaborate more on this?	As above – we have changed this to clarify that it was the principles of rehabilitation that we explored rather than the theoretical basis
Page 5: Data collection: Considering that the focus groups were done online, how did the authors ensure that the voices of all participants was heard?	Some additional detail has been added which highlights that people were given the opportunity to have further discussion following the meeting. Nobody requested this as the facilitators of the focus group are experienced and ensured that everybody was able to voice their opinion.
Page 6: Patient/Public involvement: Can you clarify exactly who were the persons consulted? Were they informal caregivers, associations/societies or persons living with dementia?	Detail has been added about the membership of the group
Page 6 Line 24: Typo: Held not had	This has been corrected
Findings Can the quotes be slightly tweaked to be more clear? For example by removing repeated words?	We have tidied up the quotes but made sure that the meaning of the quote has not been lost or altered from the verbatim transcription
Page 8: Theme: Culture: Would the term ‘organisational culture’ be more fitting here?	Yes – this has been changed. We referred to culture of the organisation, but we agree this makes the theme more clear

Page 10 Line 45-48: “The influence of the service or organisation that a person was working within, or the person with advanced dementia was being treated by, had a significant impact on the interventions that could be delivered” – This sentence is not clear. Consider rephrasing.	This has been reworded
Page 11 : Sub-theme ‘Dementia’ could be rephrased ‘dementia knowledge’	We have removed the sub headings as we believe it flows better without these (also advised by reviewer 2). We agree that this subtheme relates to knowledge of dementia
Discussion: Page 15: Line 24: Table 2 not Table 1	Corrected
Page 16: Line 6: You rightly argued that training/education alone is not enough to change the organisational culture. Can you elaborate more on factors that can support this change in culture (e.g., effective leadership, more resources, person-centred vision etc.)	Some additional text has been added
Consider including the limitations of the study. Also consider discussing some recommendations emanating from the study.	Limitations have been added
Reviewer: 2 Dr. Miia Rahja, Flinders University Comments to the Author: This qualitative study explores health care professionals’ thoughts about rehabilitation to people living with advance level dementia. The topic is interesting and the paper has potential to capture many readers, but there are some major edits that need to be done before I would recommend this for publication. The introduction does not give a clear justification for this study. The mention of this study providing a theoretical basis is ambitious and somewhat misleading and I would suggest removing any reference to this. Several of the quotations provided are oddly worded and I wonder if this is because the participants were from non-English speaking background or the transcription has not been properly checked.	Thank you for taking the time to review this paper and for your very useful comments. We have significantly re-worked the paper based on your comments. We have added more detail to the introduction to provide a clearer rationale for the study. We have changed reference to the “theoretical basis” and rephrased this throughout. Several participants had English as a second language and therefore the grammar is a little

	confusing at times. We have tidied up the quotes to avoid these being difficult to read/understand.
Abstract  - Missing detail about what methods exactly were used. It is also odd that strengths and limitations are listed in here. 	Further detail has been added here about the methods. The strengths and limitations were not meant to be in this section. Apologies – it has been moved.
 - This paper has added to the literature about HCP thoughts/knowledge/beliefs/attitudes (whatever you want to call it) about providing care for people living with advanced dementia but with a sample size of 20, it does not provide a theoretical basis. 	We agree and have removed reference to the “theoretical basis” throughout
 - The results in this sections should be reworked as well. 	This has been reworked.
Background. This section is very short and does not give clear justification to why this study is needed. My initial thought was that perhaps some of the theories that have been used to understand rehabilitation should be discussed, and why they may not suit in the context of advanced dementia. However, after reading your paper, I would suggest you remove any references to theory, or theoretical basis. This paper provides results from a qualitative study that add to other literature on the same topic and has the potential to be an interesting read. There have been other studies that discuss interviews with healthcare professionals about rehabilitation for people living with dementia, perhaps that should be the focus and how this paper is different to those studies and why this study was needed.	The background section has been reworked accordingly.
Methods  - Remove section about PPIE on page 5 as you also have this as sub-heading and section on page 6 	Removed
 - Data analysis section could give example of the narrowing of the codes to themes that are presented in this paper. 	An example has been added
Results  - Table 1 – given “other’ is just 1, could you name this profession as well. 	Added
Table 1 – how can years of clinical experience be n/a if everyone was a HCP?  - Same for % of caseload of people with dementia – maybe these should be explained in a footnote. 	One person did not want to declare their years of clinical experience – we have changed this to reflect this. The participants with N/A for % caseload were not currently working clinically and therefore did not currently have a caseload – this has been reworded to

	clarify
- Page 9, line 23 – when you state “several” can you add in brackets (n=?) so this is clear to readers – I would do this to all such statements.	We have clarified that this was discussed in all of the focus groups rather than giving specific numbers of people
Page 9 line 30 – check capitalisation	Corrected
- Page 10, line 31, - consider revising sentence beginning with: Where the acute setting was often... - Also not sure the end of this section is about determining purpose.	This has been reworded
- Page 11, line 5 – poor lighting or inappropriate lighting ?	This has been changed
Page 12 line 34 – is there a typo in this quote	This has been corrected
- Page 13, lines 55-59 – again this quote seems a bit odd as it's meaning is lost with poor grammar, for example, what does 'I have [with] the physiotherapist' mean or refer to?	The quotes have all been tidied with unnecessary text removed.
The use of subheadings under each theme makes reading of the results “clunky”. Are they necessary? Rather, focus on presenting the story that makes up each theme.	We agree and have removed the subheadings to improve the flow of the results
- Page 14, line 35 - How was it evident that your participant felt significant amount of compassion towards their patients?	Additional text has been added
Discussion Page 15, line 2 should this refer to Table 2?	Corrected
Page 15, line 27, how exactly does the data suggest that “knowledge is a key domain to ensure people with advanced dementia receive optimal care”?	Additional text has been added to clarify
Page 15 line 45 I suggest you remove terms such as “it was very evident” – it might be better to err on the side of caution especially as the paper has now clearly shown how many participants said such thing.	This has been re-worded
- It is not quite clear how you came in to the conclusion about some of the outcomes presented in Table 2. Also, how do you know that if all three “themes” from your study are poor the outcome for person with advanced dementia is “Vicious spiral of worsening outcomes”. I also wonder if (a revised version of this table – if you choose to keep it) would be better at the end of results with a summary paragraph that clearly explains it. It would be better to focus the discussion on positioning you study among other similar studies.	We have moved this table to the end of the results section and added some additional text
- Study strengths and limitations are missing, should these be moved from the abstract to here?	These have been moved
Conclusion Consider revising, would be it safer to state this study may help guide future practice in care of people with advanced dementia?	This has been revised.

VERSION 2 – REVIEW

REVIEWER	Anthony Scerri Department of Nursing, Faculty of Health Sciences , University of Malta
REVIEW RETURNED	11-May-2023

GENERAL COMMENTS	Thank you for including my suggestions in the paper and well done.
--

REVIEWER	Miia Rahja Flinders University
REVIEW RETURNED	24-Jun-2023

GENERAL COMMENTS	Thank you for the opportunity to review this revised manuscript. The authors have done a good job in addressing reviewer comments which has helped improve the manuscript and it's flow. However, there are still some comments and remarks that should be considered. Title Typo in title – professionals' Abstract Objectives – aim of the study was? You only have one aim. Results- please revise this section- it is unclear what the three overarching themes are, please state these clearly. Article summary - Remove "We acknowledge that". Also revise sentence as it is long and difficult to follow. - Provide detail how the methods were robust and what clear guideline was followed. - This summary could be stronger if more study specific detail regarding the strengths and limitations was provided. Background Thus, the importance of merging biomedical and psychosocial aspects of health, which encouraged clinicians to consider not only biological components, but also the individual and societal contexts of the individual's experiences. However, the challenge that this poses to healthcare professionals is poorly understood – especially for people living with more advanced dementia. – revise. I am surprised work by Cations or Goodwin is not cited here or in discussion. Methods/design At one stage the study refers to experts and then to specialists. Would it be better to stick to just one, specialists. Given the participants' status is unknown to readers and this can be debatable, might be safer to stick to specialist. Patient and Public Involvement (PPI) Revise (break into two) the final sentence in this section. Assuming it is meant to state that the discussion was held with representatives following the groups which helped guide the analysis and results. Also, later this group is referred to as PPIE, which one is it? Please use one term throughout for consistency. Analysis Check grammar in first sentence starting with "NVivo 11..." Same with the second sentence starting with "Familiarisation..."
---

	Findings Page 8, final paragraph... first explores how organisational culture... Same paragraph – does knowledge refer to “knowledge about dementia” or what? Could this be clarified please? Also, to make it easy for readers to know exactly what the key three themes are, could the authors clearly state these in the beginning for example: The principles of rehabilitation was explored for people with advanced dementia and three overarching themes were developed following analysis. These were ‘organisational culture’, ‘knowledge about dementia’ and ‘personal values of the HCPs’. This way there would be no room for guessing work around what exactly these themes were as they seem unclear in the current version. Organisational culture: Page 9, line 28: the impairment that resulted from what? Overall: All but one quote (OT) were from Physiotherapists. Yes most participants were physios, but it would have been also good to see quotes from the nurses and psychologists given they were participants too. Otherwise, this could be a paper about Physiotherapists’ perceptions. The use of Table 2 is still puzzling to me and not quite accurate. Would a continuum Figure be more appropriate? It’s a bit harsh to say that if ‘Knowledge of the HCP’ or the ‘organisational culture’ is poor that this automatically means that outcomes for a person living with dementia are poor as well – this cannot, and should not, be stated for sure. Discussion As per my comments in results – please provide more specific information – e.g. knowledge about what? Is it knowledge about dementia, knowledge about interventions for dementia – if both, then also please state that. Paragraph 3, It is unclear what is meant by : Lack of knowledge amongst healthcare professionals is inconsistent... inconsistent with what? Consider revising this sentence. Overall, this discussion lacks the “so what” element. What exactly can we take as a message from this study and what are the clinical implications? The strengths and limitations are only briefly mentioned now in the beginning under article summary. These are vague points, and their implications are not discussed in any detail. For example, it is clear that the participants were passionate about providing good quality care for individuals with dementia, which is a bias itself. Another limitations is that mainly physiotherapists took part in the study. There are plenty of other limitations (but also strengths) that are specific to this study but none are discussed. Conclusion I would recommend the authors provide more specifics in the conclusion. What important information does this study provide? How is this different from existing information about HCPs perceptions about rehabilitation for people with dementia? What specific new strategies would you recommend based on your data? There is plenty of data regarding HCPs perceptions about rehab for people living with dementia and the conclusion would be much stronger if the authors could discuss the novelty of their findings and how they are relevant to people with advanced dementia and HCPs treating them including how this might be used to design future studies.
--	---

I am surprised work by Cations or Goodwin is not cited here or in discussion. Methods/design At one stage the study refers to experts and then to specialists. Would it be better to stick to just one, specialists. Given the participants' status is unknown to readers and this can be debatable, might be safer to stick to specialist. Patient and Public Involvement (PPI) Revise (break into two) the final sentence in this section. Assuming it is meant to state that the discussion was held with representatives following the groups which helped guide the analysis and results. Also, later this group is referred to as PPIE, which one is it? Please use one term throughout for consistency. Analysis Check grammar in first sentence starting with "NVivo 11..." Same with the second sentence starting with "Familiarisation..." Findings Page 8, final paragraph... first explores how organisational culture... Same paragraph – does knowledge refer to "knowledge about dementia" or what? Could this be clarified please? Also, to make it easy for readers to know exactly what the key three themes are, could the authors clearly state these in the beginning for example: The principles of rehabilitation was explored for people with advanced dementia and three overarching themes were developed following analysis. These were 'organisational culture', 'knowledge about dementia' and 'personal values of the HCPs'. This way there would be no room for guessing work around what exactly these themes were as they seem unclear in the current version. Organisational culture: Page 9, line 28: the impairment that resulted from what?	guidance. The guidance does not ask for a specific summary – please inform us if this is incorrect. ] We have revised this sentence although were unclear as to which aspect you felt needed revising. To clarify, Victoria Goodwin is a co-author and we have already cited research she has been involved in but have added additional references at your suggestions. We have also cited work by Monica Cations now. It has been changed to specialists throughout This has been corrected to be PPI throughout. I feel breaking this sentence up in unnecessary and affects the flow of the text.
---	--

Overall:

All but one quote (OT) were from Physiotherapists. Yes most participants were physios, but it would have been also good to see quotes from the nurses and psychologists given they were participants too. Otherwise, this could be a paper about Physiotherapists' perceptions.

The use of Table 2 is still puzzling to me and not quite accurate. Would a continuum Figure be more appropriate? It's a bit harsh to say that if 'Knowledge of the HCP' or the 'organisational culture' is poor that this automatically means that outcomes for a person living with dementia are poor as well – this cannot, and should not, be stated for sure.

Discussion

As per my comments in results – please provide more specific information – e.g. knowledge about what? Is it knowledge about dementia, knowledge about interventions for dementia – if both, then also please state that.

Paragraph 3, It is unclear what is meant by :
Lack of knowledge amongst healthcare professionals is inconsistent... inconsistent with what? Consider revising this sentence.
Overall, this discussion lacks the "so what" element. What exactly can we take as a message from this study and what are the clinical implications?

The strengths and limitations are only briefly mentioned now in the beginning under article summary. These are vague points, and their implications are not discussed in any detail. For example, it is clear that the participants were passionate about providing good quality care for individuals with dementia, which is a bias itself. Another limitations is that mainly physiotherapists took part in the study. There are plenty of other limitations (but also strengths) that are specific to this study but none are discussed.

This has been corrected

This has been corrected

The text has been clarified.

We have added a sentence and sub-headings to aid the reader

Conclusion

I would recommend the authors provide more specifics in the conclusion. What important information does this study provide? How is this different from existing information about HCPs perceptions about rehabilitation for people with dementia? What specific new strategies would you recommend based on your data? There is plenty of data regarding HCPs perceptions about rehab for people living with dementia and the conclusion would be much stronger if the authors could discuss the novelty of their findings and how they are relevant to people with advanced dementia and HCPs treating them including how this might be used to design future studies.

References

Update reference list as there are some missing detail regarding papers which are available. Also some important references on this topic are missing.

The text has been refined.

The majority of the participants were physiotherapists. Quotes are just used to illustrate pertinent points, but data was analysed from all participants and therefore does not just represent the experiences of physiotherapists.

As a theoretical model it is simply making a suggestion about the interaction between all three factors. We have changed the title to reflect that it is theoretical which hopefully clarifies this.

Some additional text has been added.

This has been reworded to clarify

Additional text has been added

Some additional text added

This has been checked and corrected.

VERSION 3 – REVIEW

REVIEWER	Miia Rahja Flinders University
REVIEW RETURNED	13-Jul-2023

GENERAL COMMENTS	Thank you for submitting additional revisions for this paper. I only have one minor query/ suggestion regarding Table 2. If using a traffic light system, as is currently shown, should the potential outcomes highlighted in yellow be “moderate”? Thank you again for the opportunity to read this interesting paper. Well done for the authors for completing such important work.
--

VERSION 3 – AUTHOR RESPONSE

Reviewer: 2 Dr. Miia Rahja, Flinders University Comments to the Author: Thank you for submitting additional revisions for this paper. I only have one minor query/ suggestion regarding Table 2. If using a traffic light system, as is currently shown, should the potential outcomes highlighted in yellow be “moderate”?	We agree and have altered this
---	--------------------------------